# DNA repair function scores for 2172 variants in the BRCA1 amino-terminus

**Mariame Diabate**[1,2], **Muhtadi M. Islam**[1,2], **Gregory Nagy**[1,2], **Tapahsama Banerjee**[1,2], **Shruti Dhar**[1,2], **Nahum Smith**[3,4], **Aleksandra I. Adamovich**[1,2], **Lea M. Starita**[3,4], **Jeffrey D. Parvin**[1,2]*

1 The Ohio State University, Department of Biomedical Informatics, Columbus, Ohio, United States of America, 2 The Ohio State University Comprehensive Cancer Center, Columbus, Ohio, United States of America, 3 The University of Washington, Department of Genome Sciences, Seattle, Washington, United States of America, 4 Brotman Baty Institute for Precision Medicine, Seattle, Washington, United States of America

* Jeffrey.Parvin@osumc.edu

**Data Availability Statement:** Data are publicly available on GEO, accession numbers GSE116427 and GSE234482. Analysis scripts used for this analysis can be accessed at https://gitlab.com/jeffparvin/brca1_hdr_analysis.git.

## Abstract

Single nucleotide variants are the most frequent type of sequence changes detected in the genome and these are frequently variants of uncertain significance (VUS). VUS are changes in DNA for which disease risk association is unknown. Thus, methods that classify the functional impact of a VUS can be used as evidence for variant interpretation. In the case of the breast and ovarian cancer specific tumor suppressor protein, BRCA1, pathogenic missense variants frequently score as loss of function in an assay for homology-directed repair (HDR) of DNA double-strand breaks. We previously published functional results using a multiplexed assay for 1056 amino acid substitutions residues 2–192 in the amino terminus of BRCA1. In this study, we have re-assessed the data from this multiplexed assay using an improved analysis pipeline. These new analysis methods yield functional scores for more variants in the first 192 amino acids of BRCA1, plus we report new results for BRCA1 amino acid residues 193–302. We now present the functional classification of 2172 BRCA1 variants in BRCA1 residues 2–302 using the multiplexed HDR assay. Comparison of the functional determinations of the missense variants with clinically known benign or pathogenic variants indicated 93% sensitivity and 100% specificity for this assay. The results from *BRCA1* variants tested in this assay are a resource for clinical geneticists for evidence to evaluate VUS in *BRCA1*.

## Author summary

Most missense substitutions in *BRCA1* are variants of unknown significance (VUS), and individuals with a VUS in *BRCA1* cannot know from genetic information alone whether this variant predisposes to breast or ovarian cancer. We apply a multiplexed functional assay for homology directed repair of DNA double strand breaks to assess variant impact on this important BRCA1 protein function. We analyzed 2172 variants in the amino-terminus of BRCA1 and demonstrate that variants that are known as pathogenic have a loss of function in the DNA repair assay. Conversely, variants that are known to be benign are

**Funding:** This work was supported by NIH R01 CA228083 to J.D.P. and L.M.S. M.D. was supported by R01 CA228083-01A1S1. G.N. was supported by a Pelotonia Training Award. The funders had no role in study design, data collection and analysis, decision to publish, or preparation of the manuscript.

**Competing interests:** The authors declare no competing interests.

functionally normal in the multiplexed assay. We suggest that these functional determinations of BRCA1 variants can be used to augment the information that clinical cancer geneticists provide to patients who have a VUS in *BRCA1*.

## Introduction

Women with a family history of breast or ovarian cancer are encouraged to undergo genetic screening for a panel of genes including *BRCA1* (MIM: 113705) [1,2]. Germline mutations in *BRCA1* can lead to aggressive forms of advanced breast cancer, and carriers have up to a 72% lifetime risk of cancer onset [1–4]. When a variant in *BRCA1*, or any gene, is detected, it can be classified as: pathogenic, benign, or variant of uncertain significance (VUS) [5]. Due to the rarity of most variants in the general population, the majority of the variants detected are VUS. For example, in the ClinVar database [6], which records variant clinical significance, 77% of the single nucleotide variant (SNV) missense changes currently reported for *BRCA1* are VUS. The American College of Medical Genetics and Genomics advises clinicians to not give clinical recommendations for VUS [5,7].

A potential solution to this VUS information gap is found through the development and use of functional assays to measure the impact of a specific missense change on BRCA1 function. A strength of these assays is they are not reliant on aggregation of data points solely from different populations. In the case of BRCA1, previous studies have suggested that its function in homology directed repair of DNA double-strand breaks is key to its tumor suppressor activity [3,8–11]. Cells with a loss of BRCA1 function in homology directed repair assays exhibit heightened sensitivity to DNA-damaging agents and PARP inhibitors [12–15]. The use of multiplexed approaches to analyze many hundreds to thousands of variants at once [3,8,9,12,16] enable the analysis of many variants that are rare in the population.

We previously published a multiplexed homology-directed DNA repair (HDR) assay to assess function of BRCA1 missense variants on protein function [17]. Specifically, we assessed function for 1056 variants in the amino terminus of BRCA1, including the RING domain, which is known to be important in the BRCA1-mediated DNA repair activity [18–21]. After publication of that study, new standards for multiplexed functional analyses of genetic variants were published [22,23], and the approach needed to be changed to fit the new framework. We have thus modified the data analysis [12] to improve the accuracy and the number of variants for which functional interpretations can be made when using the same primary data. In this study, we re-analyze multiplexed results for the function of *BRCA1* variants in codons 2–192, and we analyze the functional effects of variants in codons 193–302, thus adding to the BRCA1 protein residues evaluated for function. These new results more than double the number of variants in the BRCA1 amino terminus for which we have functional results in DNA repair and are consistent with the results of other functional assays and with variants with known impacts in clinical predisposition to breast and ovarian cancer. Further, these new results give added insights into the biological function of amino acid residues of the BRCA1 protein in DNA repair.

## Results

### Generation of updated functional scores for BRCA1 amino-terminal variants

We assessed variants of codons 2–302 in the 1863 residue BRCA1 protein for function in multiplexed DNA repair assay. These codons were analyzed in three pools of approximately 100

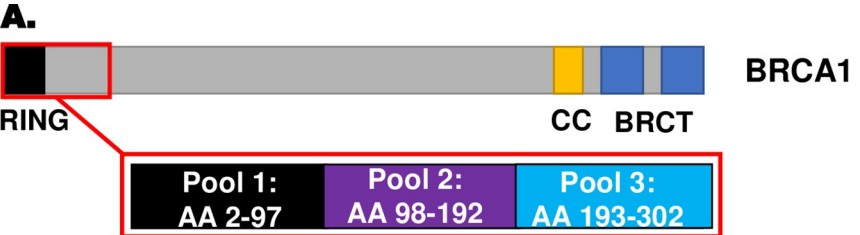

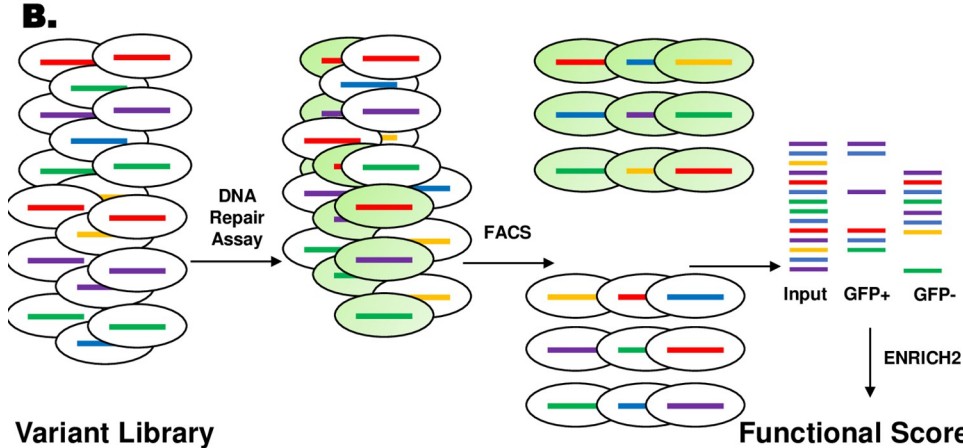

**Fig 1. BRCA1 protein and the outline of the multiplexed assay for DNA repair function of missense substitutions.**
**A.** The 1863 amino acid BRCA1 protein was evaluated for function using a multiplexed assay for homology directed repair of DNA double-strand breaks. The amino terminus of BRCA1, containing the RING domain, was divided into three pools of approximately 96 amino acid residues each, and each pool was assayed in four separate replicates of the HDR assay. **B.** Experimental workflow for the multiplexed HDR assay. A HeLa-derived cell line (HeLa-DR-FRT) was transfected with a library of plasmids expressing BRCA1 variants. In each cell a single variant of BRCA1 is integrated into the single Flp-In recombination target sequence in the genome of each HeLa-DR-FRT cell. Each dish of HeLa cells contains hundreds of BRCA1 variants present in each pool. The cells are then depleted of the endogenous BRCA1 by siRNA transfection and then subjected to the HDR assay. Cells with a variant of BRCA1 that maintains function will become GFP-positive in the assay, and LOF variants remain GFP-negative. By the use of flow cytometry, cells are sorted into GFP-positive and GFP-negative, and the barcode adjacent to the integrated BRCA1 variant is PCR amplified. These barcodes are then sequenced, and the abundance of each variant in each pool is determined by the abundance of the variant-associated barcode in the sequence reads. The abundance of a barcode representing a variant is compared to the abundance of the barcode representing the wild-type BRCA1, and analyzed using the Enrich2 software to generate a functional score.

codons, as indicated in Fig 1A. The first two pools (codons 2–192) had previously been analyzed [17]. Since the time of that finding, the analysis method has been modified [12] and the data from BRCA1 codons 2–192, along with previously unpublished results from codons 193–302, have been reanalyzed using the new analysis pipeline.

A multiplexed plasmid library containing variants of BRCA1 amino-terminal residues were generated using established methods [8,12,24,25] and previously described [17]. The plasmids in the library for the expression of missense variants were each labeled with a unique barcode, and barcodes were linked to the missense variant by long read sequencing. The plasmid library was integrated into a modified HeLa-DR-FRT cell line that contained a single Flp-In Recombination Target (FRT) sequence; each cell therefore had a single BRCA1 missense variant in its genome. For cells that were transfected with the control siRNA and analyzed for HDR

function, the endogenous wild-type BRCA1 protein was present, and all cells should be functional for HDR function. For cells transfected with the BRCA1-3'UTR siRNA, the endogenous BRCA1 would not be expressed, and the cells would be dependent in the assay on the BRCA1 variant integrated in the FRT site. Cells competent for HDR convert to GFP-positive and could be separated from the GFP-negative cells by flow cytometry. Barcode sequences were amplified from the genomic DNAs in the GFP-positive and GFP-negative pools, and based on the abundance of a variant in each pool a functional score was determined (Fig 1B). The functional score for wild-type BRCA1 was set at a value of 1, and variants would be expected to have a score between 0 and 1. Since the scores were $\log_2$ transformed, wild-type function would have a final value of 0 and loss of function variants would have a negative value.

For both, the previously published [17] and the current analytic approaches, the same sequenced library files and Enrich2 protocol were used. The major differences between analysis protocols were in the methods used to filter the data. As shown in S1 Fig, data were analyzed in two steps: first we removed the data points for which the abundance of variants in the sequenced library was too low to yield reliable results. The second step is to then determine the threshold for functionally normal versus loss of function (LOF). The differences in the analytic approaches for these two steps of the analysis are outlined in S1 Fig. The top portion of S1 Fig compares the differences in how variants with low numbers of reads were filtered, and the bottom portion of S1 Fig compares how the functional impact of the variant was determined.

In the previous analysis, low abundance variants that would potentially yield spurious results were filtered out by setting a read count threshold using false discovery rate (FDR) and adjusted q values to determine if a variant was depleted from the functional pool (GFP-positive) in the control reaction. There were four replicates of each experiment, and variants that did not pass the read count threshold in three or more of the replicates were discarded. The functional interpretation was then based on the sum of the replicates in which the q-value for a variant in the BRCA1 siRNA experiment indicated it was depleted. If a variant was scored as depleted in three or four replicates then it was scored as LOF, if a variant was depleted in no replicates, then it was interpreted as functional, and depletion in one or two replicates were not functionally interpreted (S1 Fig, *bottom left*).

In the new analysis approach, after removing variants with read-counts below the threshold (Methods), and by using the nonsense variants as internal controls for LOF, the functional score calculated by Enrich2, which represents the $\log_2$ ratio of the abundance of the variant in GFP+/GFP- populations, was directly used to interpret functionally normal versus LOF. Functionally normal (wild-type) was set at a value of 0, and functionally abnormal (complete loss of function) were less than 0.

To set the threshold for LOF we separately evaluated three populations: missense, nonsense, and synonymous variants (simulated dataset in S1 Fig, *bottom right*). The x-axis indicates the variant functional scores, and the y-axis indicates the variant counts. Under the ideal control siRNA conditions, all three populations should have a normal distribution centered on a functional score of 0, representing normal DNA repair function. Under the conditions of depletion of the endogenous BRCA1, it is anticipated that the missense variants separate into two populations, the larger population centered around 0 (functional) and a smaller peak shifted to the left on the x-axis and were LOF. The entire population of nonsense variants would be expected to be LOF and shift to the left. The synonymous variants would be expected to remain centered around the functional score of 0 (Fig 2, *bottom right*). The threshold for LOF was set as the lowest one percentile of the missense variants under control conditions, and similarly, the threshold for normal function was set as the highest one percentile of the nonsense variants in the cells depleted of endogenous BRCA1 (Methods).

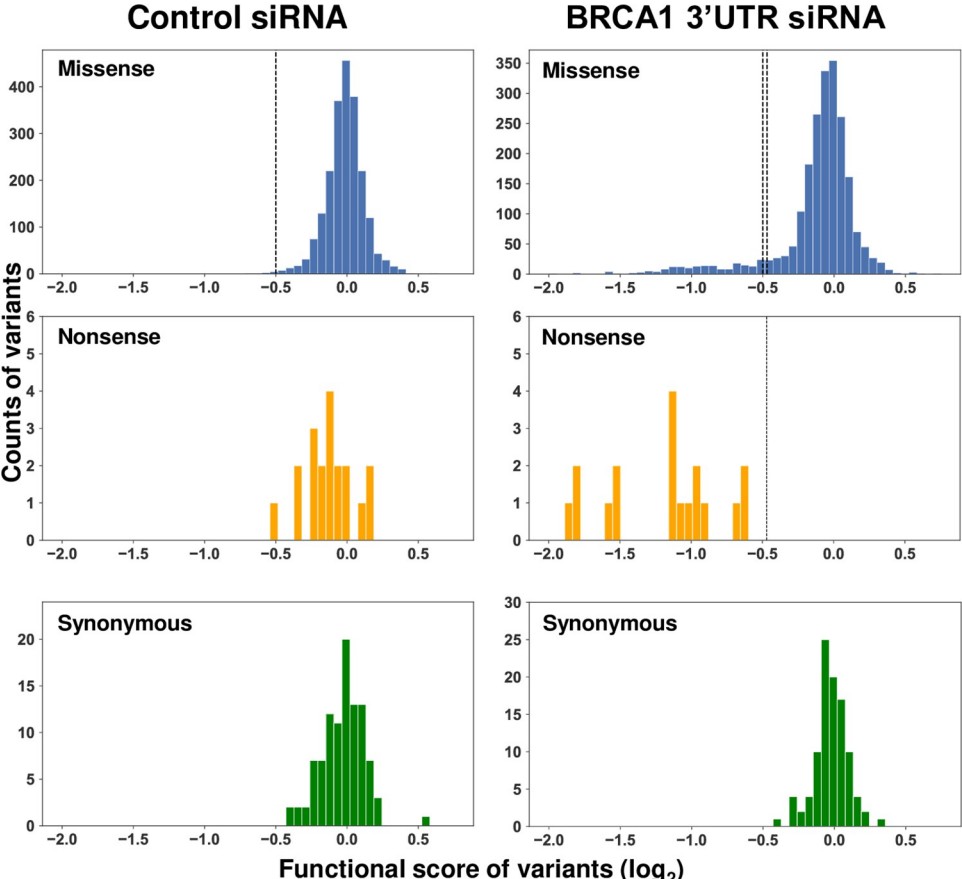

**Fig 2. Comparative population distribution of BRCA1 variants under control and experimental conditions.** The distribution of functional scores for each subpopulation of variants (missense, nonsense, synonymous) was analyzed on a plot of functional score (x-axis) vs count (y-axis). Control siRNA treatment is on the left, and *BRCA1* 3'UTR siRNA treatment (depletion of endogenous BRCA1) is on the right. The functional score scale is $\log_2$, and a score of 0 indicates wild-type function. The dotted line on the missense variant plot in the control experiment represents the bottom 1% of the normal distribution modeled on the data. The dotted line in the nonsense variant plot in the transfected BRCA1 3'UTR siRNA experiment is the top 1% of the normal distribution modeled from the data. These lines were used as thresholds for the functional interpretation of LOF.

## HDR functional assessments of BRCA1 variants in codons 2–302

For the current analysis, 2172 variants were captured (S1 Table and Fig 2). The results indicate that in cells transfected with the control siRNA (wild-type BRCA1 present), the total population of variants had a functional score centering on a value of 0, indicating maintenance of normal HDR function, as expected. By contrast, in the experiment in which the endogenously expressed BRCA1 protein was depleted by transfection of the siRNA specific to the BRCA1 3'UTR, the cell was dependent for the DNA repair function on the variant BRCA1 integrated in the FRT site. As expected, synonymous variants remained centered on a score of 0, nonsense variants shifted to the left, indicating LOF, and the missense variants were present across both distributions. The threshold cutoff scores were less than -0.50 for loss of function and greater than -0.47 for maintenance of normal function. Functional scores in the narrow range from -0.50 to -0.47 were scored as intermediate. The larger subpopulation of missense variants (1964/2154; 91%) remained centered on a functional score of 0, and a small subset of the missense variants (190/2154; 9%) shifted to the left, indicating LOF (Fig 2). The functional scores for each variant are available in S1 Table.

**A.**

|  | Variants above Read Threshold 2018 | Variants above Read Threshold 2023 | Variants with functional calls 2018 | Variants with functional calls 2023 |
|---|---|---|---|---|
| Pool 1 (2-97) | 269 | 889 | 222 | 880 |
| Pool 2 (98-192) | 790 | 870 | 718 | 868 |
| Pool 3 (193-302) | N.D. | 413 | N.D. | 413 |

**B.**

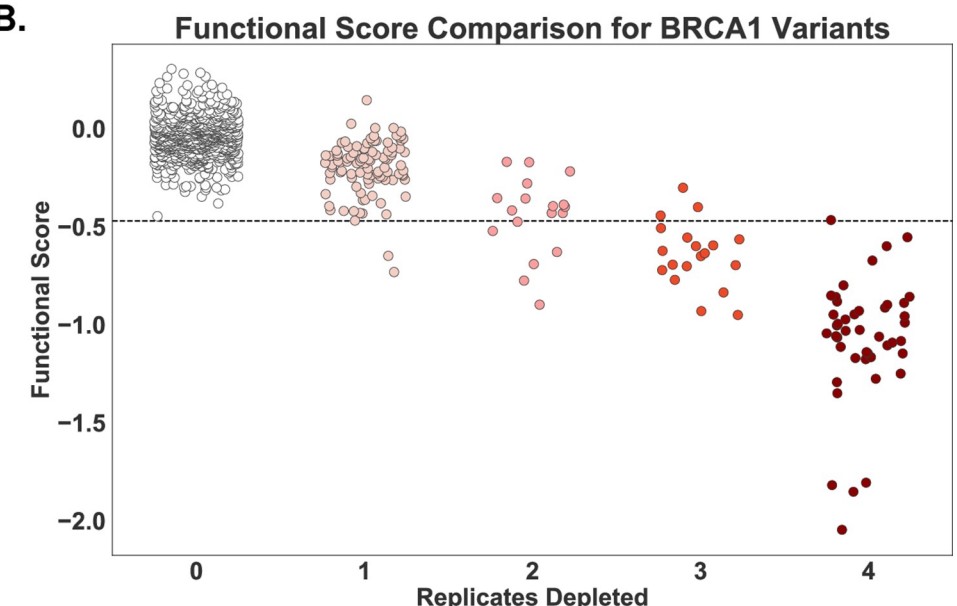

**Fig 3. Comparison of BRCA1 variant functional determinations. A.** The number of variants in each pool of *BRCA1* codons that were above the read count threshold and the number of variants for which a functional interpretation was made are indicated for the previously published analysis (2018) and the current analysis (2023). In 2018, functional determinations were only published for pools 1 and 2; only depletion scores of 0, 3 or 4 were considered for functional determination in that analysis. In the new analysis, functional determinations are made for all variants, grouped into either functionally normal, loss of function or intermediate. For this table, intermediate is not included in the total for functional calls. **B.** This scatterplot compares the functional scores for each variant from the previously published analysis (replicates depleted; x-axis) to the new analysis (y-axis). The x-axis represents the number of replicates a variant was depleted in the 2018 analysis [17]. The y-axis displays the re-analysis functional score for the same population of variants. The dotted line signifies the threshold for function, with values above it indicating functionally normal BRCA1 and below indicating loss of function. Colors indicate the color scheme used in the 2018 description. The comparison showed a strong negative correlation, with a Pearson R value of -0.86.

We compared the functional interpretation for codons 2–192 from the prior analysis and the current re-analysis. With the change in analytic approaches, we were able to make functional calls for 808 more BRCA1 variants than previously analyzed in codons 2–192 (Fig 3A). In the prior work, we assessed function for 269 and 790 variants in Pools 1 and 2 respectively, however we only made functional interpretations for 222 and 718. Currently, we were able to provide functional interpretations for 880 and 868 variants for Pools 1 and 2 respectively. The observed increase in variant coverage in Pools 1 and 2 may be attributed to the more flexible approach in determining the read-counts threshold, which allows for greater inclusion of reads in variant calling and improves overall coverage.

We directly compared the functional interpretations from both analytic approaches by plotting the replicates depleted in the prior analysis (x-axis) [14] to the functional scores for each

variant in the new analysis (y-axis; Fig 3B). There was a high concordance between the analysis approaches, with a calculated Pearson r value of -0.86 for the 1044 variants evaluated in both studies (S2 Table). Of the variants that were seen in both analyses, all of the variants with 0 replicates depleted were scored as functional using the current methods. There were four variants for which the functional interpretation was changed: in the previous analysis three were depleted in three replicates and one variant was depleted in four replicates, and in the current method three of these; BRCA1-p.I26G, BRCA1-p.I89G, BRCA1-p.L95G have scores consistent with intermediate function, and one; BRCA1-p.T97G was functional. The results for the two analysis approaches were highly concordant, but the new approach enabled twice as many variants to have functional interpretations.

## Correlation of functional interpretations to BRCA1 variants with known clinical impact

We compared the functional interpretations based on the multiplexed DNA repair assay to the clinical variant classifications reported in ClinVar [6]. For the 19 variants in the dataset classified as either benign or likely benign (S3 Table), the functional score centered on 0 under control conditions as well as when endogenous BRCA1 was depleted (Fig 4). Thus, these known benign variants were functionally normal in DNA repair. The 33 pathogenic or likely pathogenic variants were analyzed in the multiplexed DNA repair assay. Under the control siRNA the variants center around 0. For the BRCA1 3'UTR siRNA condition, 29 of the 33 variants shifted to the left to functional scores less than -0.5 (left of the dotted line), consistent with LOF. Four of the pathogenic variants were scored as functionally normal. Three of these variants BRCA1- p.R71G, BRCA1- p.R71K and BRCA1- p.R71W are known to affect splicing [26,27], and these would be missed in the HDR assay since it uses cDNA to express BRCA1 variants. The fourth pathogenic variant misclassified in these results was BRCA1- p.P34R, which had a functional score of -0.410, and this score was in the range assigned to functionally normal (>-0.47). Of the 240 variants in ClinVar classified as VUS or with conflicting interpretation, the functional scores (Fig 4, *bottom*) suggested two subpopulations: a larger subset with normal function and a smaller subset with functional scores that shifted to less than -0.5, indicating LOF. It should be noted that the ClinVar database now includes the results from functional assays in a separate category for each listing, but for the variants considered in Fig 4 we only used the clinical classification. As a caveat, it is possible that the clinical classification for a given variant in ClinVar may be informed by the functional results from previously published assays. We calculated the strength of evidence at which this functional data be applied to clinical variant interpretation workflows using the 'odds of pathogenicity' [5,7] formula provided by the ACMG in S6 Table. Based on the 52 variants with clinical classifications tested in the multiplexed HDR assay (S3 Fig), variants classified as LOF can be used as PS3 moderate and variants classified as functionally normal can be applied as BS3 strong (S6 Table).

The previously published analysis approach recovered fewer variants with known phenotype listed in ClinVar with four of the five benign variants scoring as functional. Of the pathogenic variants previously analyzed, 10 of the 11 were correctly classified as loss of function. (The numbers of pathogenic variants analyzed in the 2018 study increased using the updated ClinVar data.) The current analysis approach increases the sensitivity and specificity of functional classifications with reference to ClinVar (S3 Fig).

## Comparison of the multiplexed DNA repair assay to existing functional data for the BRCA1 RING domain

An orthogonal multiplexed assay for BRCA1 function, called Saturated Genome Editing (SGE), was used to evaluate the function of missense variants in 13 exons of BRCA1, including

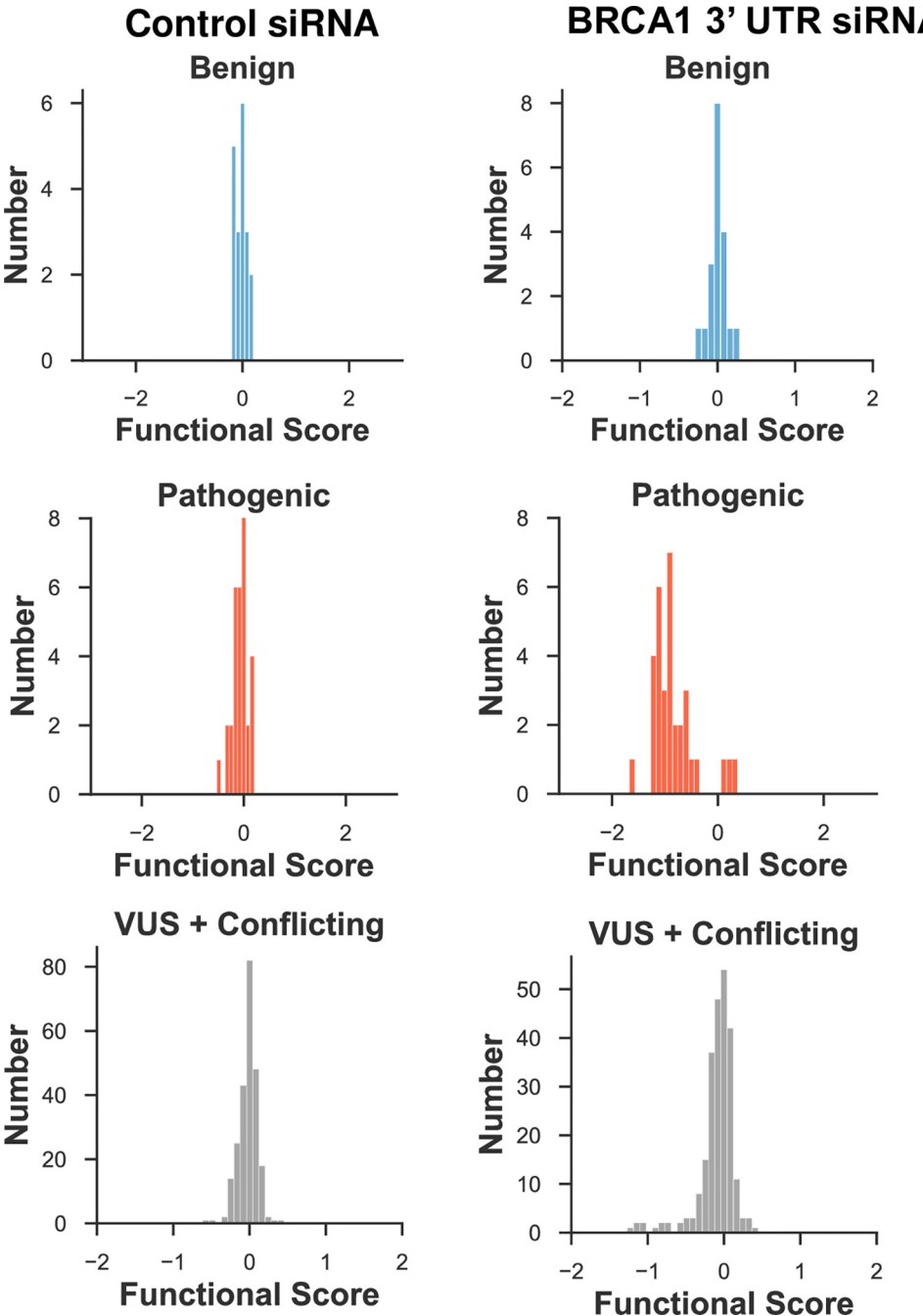

**Fig 4. Comparison of functional scores for BRCA1 variants with ClinVar classifications.** Under the conditions of endogenous, wild-type BRCA1 (control siRNA, left) or of variant BRCA1 (BRCA1 3'UTR siRNA, right), the variants present in the ClinVar database and classified as benign/likely benign (top), pathogenic/likely pathogenic (middle), or VUS/conflicting (bottom) are shown. These populations are depicted on a plot of functional score ($\log_2$, x-axis) versus number of variants (y-axis). Dotted lines added to each plot represent the functional cutoffs for the analysis. Variants to the right of the dotted line are functionally normal, and to left of the dotted line are loss of function.

the RING domain [16]. There are differences in the approaches used for the SGE assay and the multiplexed HDR assay evaluated in the current study. The SGE assay analyzed all single nucleotide variants in amino acid positions 1–100, contrasted with all possible codon

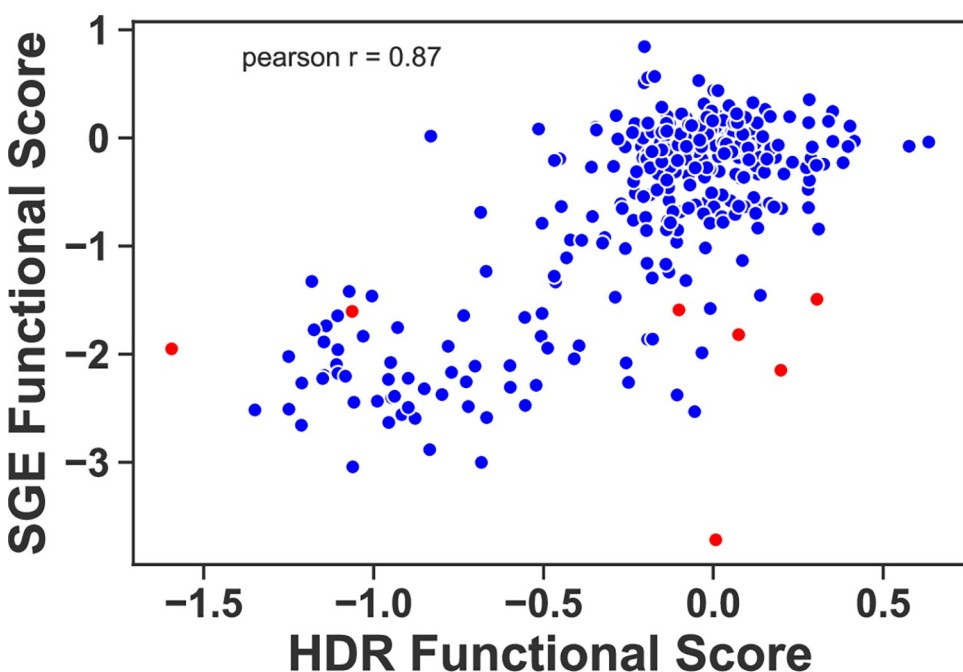

**Fig 5. Comparison of functional scores for BRCA1 variants in the HDR assay versus the Saturated Genome Editing (SGE) assay.** The functional scores from the current multiplexed HDR analysis (x-axis) were compared to the multiplexed scores from the SGE dataset (y-axis) on a scatterplot [16]. Variants colored red indicate low RNA levels as recorded in the SGE dataset. Both datasets use a logarithmic functional score scale with base 2, and a score of 0 is the functional score of wild-type.

substitutions in positions 2–302 in the current study. In addition, SGE depended on CRISPR editing of the genomic locus of *BRCA1* in a haploid cell line, and this enabled effects on splicing to be included in functional analysis. Lastly, the SGE assay is a proliferation assay, whereas the multiplexed HDR assay measures DNA repair. We compared the functional scores from the multiplexed DNA repair assay (x-axis) to the functional scores published from the SGE proliferation assay (y-axis), and we observed a strong correlation between the functional assays as indicated with a Pearson r value of 0.87 (Fig 5 and S4 Table). The variants aggregated in two groups, a major group of functionally normal variants in both assays centering on 0 on each axis, and a second smaller group representing LOF in both assays. Interestingly, many of the intermediate scores for both assays were also consistent. Of 312 overlapping variants, only 33 (11%) were discordant when comparing the two assays, and seven of these were due to variants that affect the abundance of the RNA in the SGE assay (indicated in red).

The effects of missense variants on BRCA1 have been tested in a number of studies by analyzing one variant at a time in the HDR assay–the singleton assay [8,13,28–32]. For the current study, we have analyzed 18 different variants in the singleton assay (Fig 6A) in addition to variants in the BRCA1 amino-terminus that have been previously published [8,12,17,33]. For comparing the current results to available singleton HDR results, we transformed the functional score in the singleton assay to log$_2$, as is done for the multiplexed assay and compared them on a scatterplot (Fig 6B). Loss of function for the singleton assays was less than 0.4 relative to the wild-type control (-1.322 log$_2$ transformed) and functionally normal was greater than 0.7 relative to the wild-type (-0.515 log$_2$ transformed). The results of the two assays were highly correlated (Pearson r = 0.87), indicating confidence in the current results. Of the 44 variants tested in both the multiplexed and singleton HDR assays, there were four (9%) with results that were discordant: BRCA1-p.I15L, BRCA1-p.F93A, BRCA1-p.T176K, and BRCA1-p.C226T (Fig 6B).

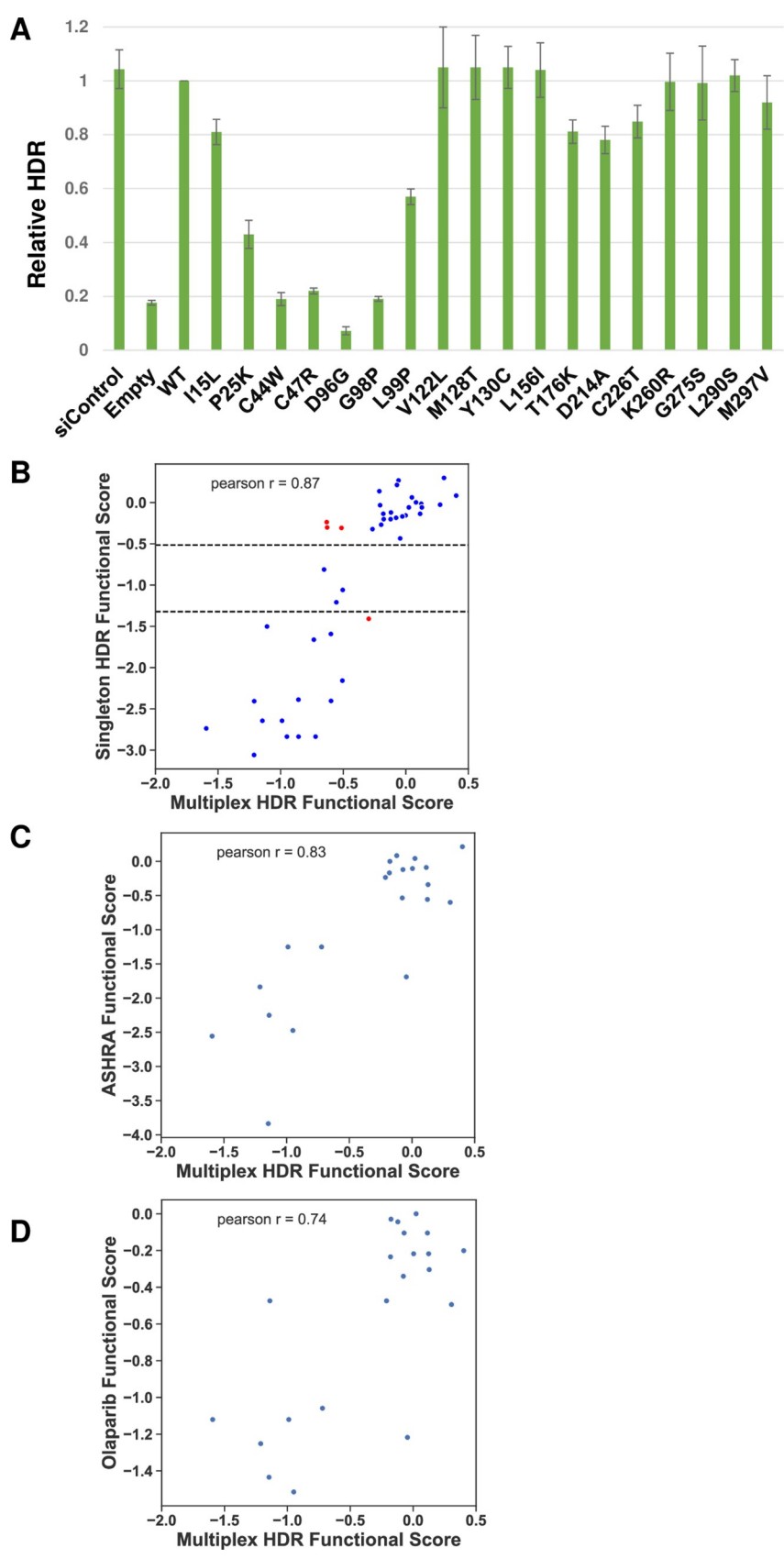

**Fig 6. Comparison of functional scores for BRCA1 variants assessed in multiplexed and singleton HDR. A.** Eighteen amino-terminal BRCA1 variants were tested in the singleton HDR assay for DNA repair function. In the first bar, the control siRNA was transfected into the cells, and in all the other samples the siRNA specific to the 3'UTR of BRCA1 was transfected. The encoded BRCA1 variant used to rescue the depletion of the endogenous BRCA1 was indicated. **B.** The functional assessments of 44 BRCA1 variants (18 from panel A and 26 published) was compared in both the HDR multiplexed assay (x-axis) and the HDR singleton assay (y-axis). The dotted lines indicate the functional cutoffs for the singleton assay, with functionally normal defined as a score of > -0.51 and non-function defined as < -1.32. The comparison showed a strong positive correlation, with a Pearson R value of 0.87. Both datasets were $\log_2$ transformed. Discordant data points are colored in red. **C.** The functional scores from the current multiplexed HDR analysis (x-axis) were compared to the singleton scores provided from the ASHRA assay (y-axis) [36]. The functional scores for the singleton assay were $\log_2$ transformed. **D.** The functional scores from the multiplexed HDR analysis (x-axis) were compared to the singleton scores for Olaparib resistance [36]. The functional scores for the singleton assay were $\log_2$ transformed.

We also compared our results (S6 Table) to the singleton variants tested using a different assay for homology directed repair (Fig 6C, Pearson r = 0.83) and an assay for resistance to Olaparib (Fig 6D, Pearson r = 0.74) [13]. Both datasets were $\log_2$ transformed and the results from the assays were highly correlated.

## BRCA1 residues required for HDR function

The sequence function map for BRCA1 2–302 (Fig 7) shows that 181/183 of the loss of function variants we classified were found in the RING domain (AA 1–109). The two LOF variants

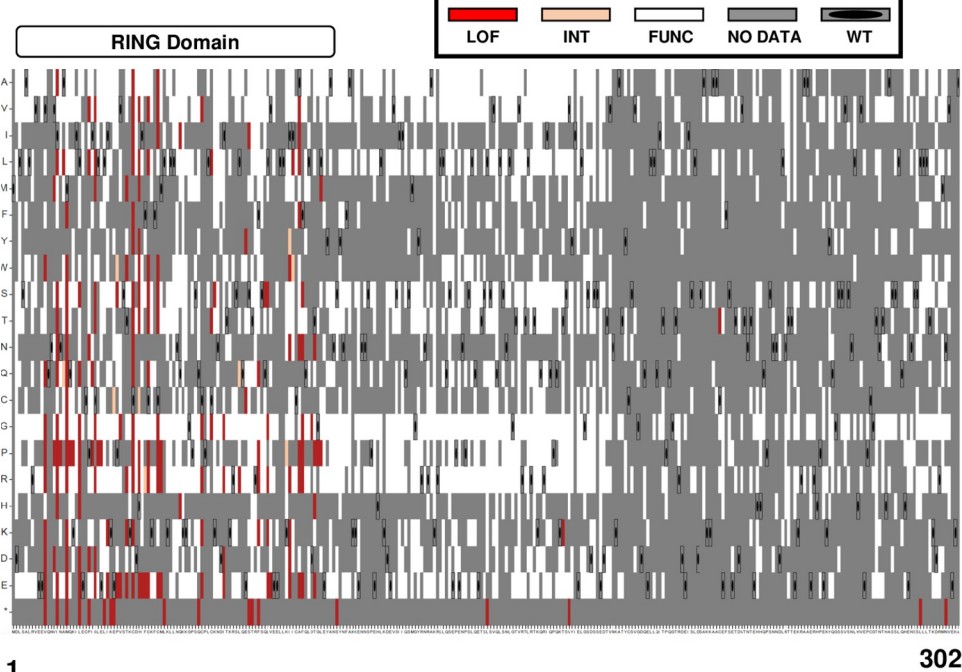

**Fig 7. Sequence-Function map of 2172 BRCA1 variants.** The functional scores of 2172 variants (positions 1–302 of BRCA1) were color-coded based on the functional scores for each variant. The x-axis has wild-type BRCA1 amino acid residues from 1 to 302, and the y-axis has the amino acid substitution indicated. Substitutions that resulted in the wild-type amino acid being generated were indicated by a grey box containing an oval dot. Grey indicates variants for which the read counts were below the threshold for inclusion or where a functional score was not reported. Variants with a functional score less than -0.5 were colored red, indicating loss of function (LOF). Variants with a functional score greater than -0.47 were colored white, indicating maintenance of function. Intermediate function variants with scores in between were colored peach. Nonsense variants were marked with an asterisk (*) on the bottom row of the y-axis.

found outside of the RING domain (BRCA1-p.T176K, BRCA1-p.C226T) were functional in the singleton DNA repair assay (Fig 6A and 6B), and one of these two LOF variants (BRCA1-p.T176K) has been predicted as benign in silico [34,35]. Since these two multiplexed assay results were likely spurious, we conclude that BRCA1 residues required for HDR function, and which are therefore inflexible to substitution, reside solely within the RING domain. The sequence function map of just the RING domain is shown (S4A Fig) to facilitate evaluating individual codons.

Previously published assays that depend on replication fork stability [36] or single-strand annealing repair of DNA double strand breaks [31] had found residues of functional importance in the BRCA1 residues between 100 to 300. Residues that were loss of function in the replication fork stability assay (BRCA1-p.S114A, BRCA1-p.R133C, BRCA1-p.Y179C, and BRCA1-p.S265Y) were scored as functionally normal in the multiplexed HDR assay in this study. BRCA1 variants, BRCA1-p.I124V and BRCA1-p.V191D, were both LOF in the SSA-based assay and were found to be functionally normal in the multiplexed HDR assay.

To investigate the zinc finger RING domain residues inflexible to substitution, we examined the 3D structure for the BRCA1 in complex with BARD1 (PYMOL:1JM7) [37]. As expected, residues that coordinate the zinc atoms were inflexible to substitution (S4B Fig). Similarly, many of the BRCA1 residues in the BARD1-binding interface were inflexible to substitution, as indicated by the red and pink colored residues. Of interest, not all residues that were intolerant to substitution could be explained by binding to BARD1 or to zinc atoms. The results of the analysis showed that the residues partially intolerant to substitution, including BRCA1-p.Met-48, BRCA1-p.Gln-54, BRCA1-p.Gln-60, BRCA1-p.Ile-68, and BRCA1-p.Leu-73, are not directly involved in the formation of the BRCA1-BARD1 complex or stabilizing zinc atoms. It is likely that other protein-protein interactions involving these residues impact BRCA1 DNA repair function.

Multiplexed functional assays yield a high density of functional results for substitutions at each amino acid residue, and these substitutions may be useful in interpreting whether a defined protein motif is important in the process being tested. As an example, the nuclear export sequence, NES2, overlaps significantly with the BARD1 interaction domain of BRCA1. The NES2 motif [$^{81}$QLVEE**L**LKI**I**CA**FQL**DTGL$^{99}$] [21,38–44], had many residues tested in the DNA repair assay (S4B Fig). Six of nine substitutions tested for BRCA1-p.Phe-93 were LOF, but this residue is also part of the BARD1 binding interface. By contrast, the BRCA1-p.Leu-95 residue is essential to the NES2 motif but all eight substitutions of this Leu were functional in DNA repair. Similarly, Ile-90 is an important residue in the NES2 motif, and we found that 11 of the detected substitutions at this residue were functional in DNA repair. Some substitutions of residues of NES1 [$^{22}$**L**ECP**I**C**L**E**L**$^{30}$] [38–46] also resulted in LOF, and these could not be explained by BARD1 binding or zinc atom binding. The first leucine residue in NES1 is key to its nuclear export function [44] and substitutions resulting in loss of function. This sighting is consistent with the results in our assay for BRCA1-p.Leu-22, nine out of seventeen substitutions resulted in loss of function. However, other residues previously identified as critical for NES1 activity were functional when substituted. As an example, we assayed nine missense substitutions of BRCA1-p.Leu-28, and eight were functional and only the substitution to proline resulted in loss of function. Similarly, BRCA1-p.Leu-30 is important for NES1 activity, and we assayed 15 missense substitutions at this residue, and all but one substitution resulted in functional BRCA1 proteins. Previous studies [38,40–44] suggested that the BRCA1 NES plays an important role in regulating HDR function of DSBs in the cell nucleus, but the mechanism was unknown. The results from this study suggest that the NES does not significantly regulate HDR function, but there are many residues in the BRCA1 amino-terminus that do affect HDR in an as yet undefined way.

## Discussion

In this study, we applied a new analysis approach to previously published results from a multiplexed analysis of the function in DNA repair for BRCA1 variants in the amino terminus of the protein. Using this new approach for BRCA1 residues 2–192, we determined functional scores for 808 additional variants. Plus, we analyzed 413 variants in BRCA1 residues 193–302 that had not been previously evaluated. The new approach differed from the previously published analytic pipeline in two key steps: first, in judging whether the data for a specific variant was abundant enough in the population to be included in the results without spurious classifications (the read count threshold), and second, in interpreting LOF versus functionally normal for a given variant. In the prior study, we had evaluated the results using a binary depleted versus not depleted in the functional pool. This method was stringent but excluded the interpretation of the results from many variants that would be interpreted as functional with the current methods. In the current study, we instead calculated a functional score from the abundance of a variant in the GFP-positive pool of cells under test conditions, compared to the abundance of the variant in the GFP-positive pool of cells under the control conditions. By using synonymous variants and nonsense variants present in the dataset, we were able to treat the variants as a population distribution and identify those variants for which the functional score shifted to low values when comparing control (endogenous BRCA1 present) to test (endogenous BRCA1 depleted) conditions. This new analytic approach enabled the interpretation of the functional impact of amino acid changes for more variants than had been possible with the previous approach, and the new approach yielded results that had higher sensitivity and specificity when compared to variants with known clinical interpretation. The new analytic approach yielded some variants with a functional score in the intermediate range. The impact of an intermediate level of function is difficult to interpret since such a variant may be only partially compromised in function but whether it is disease predisposing is unclear.

One of the biggest obstacles for the previous approach was minimizing noise in our results. For multiplexed studies, it is important to separate between a true signal and stochastic noise. This problem primarily impacts the data at low read counts or replicate experiments with high variability. In the new analysis approach, we utilized empirical data from the synonymous and nonsense variants that improved the identification of the read-count threshold. In addition, we removed variants that had a mean variance greater than 1 across the four replicates. With the improved thresholding, the results we obtained for the multiplexed HDR assay were very similar to the results from the variants tested in the singleton assay. Comparing our results to variants with known clinical impact, as indicated in ClinVar, our accuracy for sensitivity and specificity was 93% and 100%, respectively. In the multiplexed DNA repair function dataset, there were 240 variants classified as conflicting interpretations or VUS in ClinVar, for which the experiments now give functional evidence for re-interpretation. Of the 240 VUS, 226 were functionally normal and 14 of them were LOF.

The inflexibility to substitution of key residues in the RING domain is supported with the results of our DNA repair assay. The high resolution of the multiplexed functional assay can be used to finely dissect whether a known domain of the protein is important for a defined activity of the protein. Not only do the functional analysis of BRCA1 variants yield functional calls that can provide interpretation of VUS, but also the high-resolution nature of the assay yields insights into the biological activities that are important in the DNA repair in the cell.

## Materials and methods

### Preparation and generation of multiplexed variant library

The variant libraries representing variants in pool 1 (BRCA1 residues 2–96) and pool 2 (BRCA1 residues 97–192) were the same as previously published [17]. The same methods were

applied to generate the multiplexed pool 3 (BRCA1 residues 193–302). The methods of integration of the plasmid library into HeLa-DR-FRT cells, HDR assay, sorting of cells, gDNA preparation, barcode amplification and sequencing were as previously described [17].

## BRCA1 variant library construction and HDR analysis

Inverse PCR reactions of the plasmid encoding the BRCA1 cDNA generate full length product, but one of the codons in the PCR oligonucleotide contains NNK (where N = A, C, G, T and K = G, T) in place of the terminal codon encoded by the oligonucleotide. For the 301 codons analyzed in this study, 301 different inverted oligonucleotide pairs were separately prepared and subjected to PCR. The PCR reactions were then pooled and circularized by the action of DNA ligase. An oligonucleotide containing a 16 bp degenerate oligonucleotide barcode was inserted upstream of the BRCA1 coding sequence, and long-read sequencing using a PacBio Sequel paired each barcode with a different BRCA1 variant. The plasmid libraries for pools 1, 2, and 3 were each transfected along with the FLP-In recombinase into a HeLa derived cell line that contained in its genome a single FLP-In Recombinase Target (FRT) sequence [45,46] and a single DNA sequence for measuring homology directed repair (HDR) of DNA double-strand breaks [47,48]

The HDR assay is initiated by transfecting a plasmid encoding the I-SceI endonuclease, which generates a DNA cut in one of the defective GFP sequences, and if the cell is competent for HDR, recombination repairs the defective GFP coding sequence, and the cell becomes GFP-positive (Fig 1B). We performed this HDR assay under two conditions. In the first condition, we have transfected a control siRNA, and the cells contain endogenously expressed wild-type BRCA1. In the second condition, we have transfected an siRNA targeting the 3'-UTR of the endogenously expressed BRCA1 mRNA, and the cell is then dependent on the BRCA1 variant expressed from the FRT site. GFP-positive cells were separated from GFP-negative cells using flow sorting, and the barcode in the genome of the cells was isolated by PCR and analyzed by short read sequencing. The frequency of the barcode in the GFP-positive cells (functional for HDR) is compared under the conditions of depletion of the endogenous BRCA1 protein using the siRNA targeting the BRCA1 3'-UTR to the condition of the control siRNA. From the abundances of the barcodes in the various samples, we calculated a functional score using the program Enrich2 [49].

## Variant scoring, classifications and depletion score

The original output files [17] in.h5 format generated by Enrich2 [49] were utilized for pools 1 and 2. For pool 3, the FASTQ files containing barcode variants and a barcode map were processed using Enrich2 software. Enrich2 calculates the functional score for each variant from its abundance in the GFP-positive and GFP-negative pools. The functional score is obtained by normalizing the slope of the line connecting the control and experimental conditions for each pool by the wild-type adjusted by log ratio. The functional score was reported in $\log_2$ for each variant tested with a value of 0 representing normal wild-type function [49].

The process of interpreting variant scores involved two steps. First, two subpopulations of variants containing synonymous and nonsense changes were identified and analyzed in a scatter plot to display the relationship between the variant read count and their functional scores for each subpopulation (S1 Fig, *top right*). Under control conditions, the endogenous BRCA1 present in every cell functioned normally and was used as a baseline for the DNA damage repair assay. Thus, it was anticipated that all variant scores in this condition would cluster around 0 on the scatter plot (S1 Fig, *top right*). The minimum-read count threshold was determined to be the highest point where the results for any variant deviated from this functionally normal value (examples circled in red on the figure). To confirm this threshold, using the

scatter plot from the conditions in which the endogenous BRCA1 was depleted by transfecting siBRCA1-3'UTR, nonsense variants were expected to exhibit negative functional scores indicative of loss of function, and the read-count threshold would be higher than any nonsense mutants that scored as functionally normal (S1 Fig, *top right*). If the read count values from the conditions were different, we selected the higher threshold. Data with lower read counts than this empirically defined value were discarded. To further refine the data, the mean variance across four replicates was calculated, and variants with variance greater than 1 were removed from the analysis (S2 Fig). These two filters removed variants with low frequency of read counts and high variability of results. The remaining variants have high frequency of datapoints that were relatively consistent in score, and we used these to evaluate functional effects of the amino acid change for each variant.

After applying the minimum reads-count threshold, a histogram was generated to visualize the distribution of the variant library (S1 Fig, *bottom right*). Three subpopulations, missense, nonsense, and synonymous, were analyzed separately for each condition. To determine the functional score thresholds, the population of missense variants in the control siRNA experimental condition was modeled as a Gaussian distribution, and the lowest 1% of values were selected as the threshold for loss of function. Similarly, using the distribution plot for the BRCA1 siRNA condition, the population of nonsense variants were modeled as a Gaussian distribution and the highest 1% was defined as functionally normal. Only variants that were observed in at least three replicates were included in the analysis. The scripts utilized in this analysis, which were developed using Python and R Studio, are available at https://gitlab.com/jeffparvin/brca1_hdr_analysis.git.

The single nucleotide changes in the BRCA1 cDNA were provided in S9 Table. Since the variant BRCA1 proteins were expressed from a cDNA construct and from the FRT site inserted in the genome, the effects that nucleotide changes have on RNA processing would not apply to the BRCA1 variants tested in this assay. The amino acid change was of primary importance in this analysis.

## Supporting information

**S1 Fig. Comparison of analysis approaches for evaluating BRCA1 variants in the RING domain.** The left side of the figure shows the original analysis steps used in the paper published in 2018 [17], and the right side of the figure summarizes the steps changed in the current approach. The analytic pipeline previously described used a binary classifier based on the false-discovery rate (q-value) as a quantifier. The binary classifier was created by designating variants with a q value <0.055 as 'depleted' and variants with a q value > 0.055 as 'not depleted.' The overall depletion score was calculated by counting the number of times a variant was depleted across the four replicates. In the current study, performance was optimized using internal controls (synonymous and nonsense variants) in cells containing endogenous BRCA1 (control siRNA) and in cells with the endogenous gene silenced (BRCA1 siRNA). We evaluated the read counts (horizontal axis) and at low read counts the datapoints deviated from normal function (0 on the vertical axis) in control cells and in the BRCA1 siRNA transfected cells, synonymous variants deviated from normal function at low read counts. The red circles indicate data for variants which deviate from expectations and were used to establish the read-count threshold. This analysis set the minimum number of reads required for a variant to be included in the analysis. After establishing the read-count threshold, the threshold for functional versus LOF was determined. In the previously published analysis, if the q-value for a variant indicated depleted in three or four replicate experiments, then the variant was considered LOF. If the q-value indicated zero replicates depleted, then it was interpreted as functional. If a variant was depleted in one or two replicates, then no functional determination was made. In

the current analysis, the population distributions of missense, nonsense, and synonymous, shown here as expected distributions, were used to determine the threshold for functional interpretation. The cut-off values were established based on the top 1% for nonsense variants and the bottom 1% for synonymous variants.
(TIF)

**S2 Fig. Mean variance of BRCA1 across four replicates.** The DNA repair functional score variability of BRCA1 variants was evaluated by plotting the standard variance across four replicates (y-axis) against the mean functional score (x-axis). Variants with a standard deviation greater than 1 were removed from further analysis.
(TIF)

**S3 Fig. Calculation of sensitivity and specificity for BRCA1 variant functional scores. A.** The current analysis of the multiplexed HDR assay was compared with variants with known clinical impact listed in ClinVar. **B.** The functional determinations using the previously published analysis was compared with variants with known clinical impact listed in ClinVar. Due to updates in the ClinVar database, the number of variants shown in this table is different from originally published.
(TIF)

**S4 Fig. Sequence-function map in the BRCA1 RING domain. A.** The close-up view of the sequence-function map from Fig 7 shows the relationship between the sequence of the RING domain (positions 1–110 of BRCA1) and the functional impacts of each variant tested. The color coding of the variants represents the functional performance of the RING domain: red for loss of function, white for functionally normal, peach for intermediate function, and gray for variants with no determinations (read-counts below the threshold for inclusion or variant not detected). The x-axis represents the wild-type amino acid one-letter code, and the y-axis represents the mutated amino acid one-letter code. **B.** This visualization shows the interaction between BRCA1 and BARD1 proteins (PYMOL:1JM7), with BRCA1 residues colored based on their performance in the functional assay. Red represented loss of function in all substitutions, light pink represented more than half of substitutions resulting in loss of function, peach for less than half of substitutions resulting in loss of function, and white for maintenance of function in all tested substitutions. The zinc atoms in the RING zinc-finger are colored grey. BARD1 peptide was colored green. In the close-up view of the alpha-helices of BRCA1, the nuclear export sequences are indicated with brackets and arrows, and the helices have been rotated to show the face of BRCA1 that interacts directly with BARD1.
(TIF)

**S1 Table. New BRCA1 Multiplexed Functional Scores and Additional Outputs.**
(XLSX)

**S2 Table. Comparison of Amino-Terminus Functional Scores for 2018 Analysis and Current.**
(XLSX)

**S3 Table. Comparison of Current Approach Functional Scores with Reported BRCA1 Variant Clinical Classifications on ClinVar.**
(XLSX)

**S4 Table. Comparison of Functional Scores and Interpretations between Current Analysis Approach and Saturated Genome Editing (SGE) Analysis.**
(XLSX)

**S5 Table. Scoring for 18 New BRCA1 Singleton Variants.**
(XLSX)

**S6 Table. Comparison of Functional Scores and Interpretations between Current Analysis Approach and Assay of Site- Specific HR Activity (ASHRA).**
(XLSX)

**S7 Table. Odds of Pathogenicity Calculations for the Current BRCA1 Variant Functional Classifications.**
(XLSX)

**S8 Table. Primer Used to Generate BRCA1 Singleton Variants.**
(XLSX)

**S9 Table. Single Nucleotide Substitution Changes.**
(XLSX)

## Author Contributions

**Conceptualization:** Mariame Diabate, Aleksandra I. Adamovich, Lea M. Starita, Jeffrey D. Parvin.

**Data curation:** Mariame Diabate, Muhtadi M. Islam, Gregory Nagy, Tapahsama Banerjee, Shruti Dhar, Nahum Smith, Aleksandra I. Adamovich, Lea M. Starita.

**Formal analysis:** Mariame Diabate.

**Funding acquisition:** Jeffrey D. Parvin.

**Investigation:** Mariame Diabate, Muhtadi M. Islam, Gregory Nagy, Tapahsama Banerjee, Shruti Dhar.

**Methodology:** Mariame Diabate, Nahum Smith, Aleksandra I. Adamovich, Lea M. Starita.

**Project administration:** Jeffrey D. Parvin.

**Resources:** Jeffrey D. Parvin.

**Software:** Mariame Diabate, Lea M. Starita.

**Supervision:** Lea M. Starita, Jeffrey D. Parvin.

**Visualization:** Mariame Diabate, Jeffrey D. Parvin.

**Writing – original draft:** Mariame Diabate.

**Writing – review & editing:** Mariame Diabate, Jeffrey D. Parvin.

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
