## [Decision Letter · Decision Letter 0]

17 May 2023

Dear Dr Parvin,

Thank you very much for submitting your Research Article entitled 'DNA Repair Function Scores for 2172 Variants in the BRCA1 Amino-Terminus' to PLOS Genetics.

The manuscript was fully evaluated at the editorial level and by independent peer reviewers. The reviewers appreciated the attention to an important topic but identified some concerns that we ask you address in a revised manuscript.

We therefore ask you to modify the manuscript according to the review recommendations. Your revisions should address the specific points made by each reviewer.

Yours sincerely,

Anne O'Donnell-Luria, MD, PhD

Academic Editor

PLOS Genetics

David Kwiatkowski

Section Editor

PLOS Genetics

Thank you for submitting your manuscript to PLOS Genetics. The reviewers agree that this manuscript is a nice addition to the field but do not several concerns that need to be addressed, particularly the circularity of using ClinVar classifications that use the Findlay et al functional data that need to be adjusted for in the results.

I would also like to echo reviewer 3's request that the data be deposited to MAVE, as this will hopefully help it to have greater reach for other researchers.

Reviewer's Responses to Questions

**Comments to the Authors:**

Reviewer #1: BRCA1 is a breast and ovarian cancer-specific tumor suppressor protein, and its homology-directed repair (HDR) activity is known to be important for its function as a tumor suppressor. Therefore, evaluating BRCA1 variants' function is critical for accurately diagnosing hereditary cancer. The authors previously published the results of a multiplexed HDR assay for 1056 amino acid (aa) substitutions in the amino-terminal region of BRCA1, 2-192 aa. In this study, the authors improved the analysis pipeline. They re-analyzed the variants in 2-192 aa and added the results of the new variants in 193-302 aa of BRCA1. Thus, they analyzed 2172 variants and improved the sensitivity and specificity for this assay compared with the determination by ClinVar, which considers clinical information.

This study is novel and interesting and will contribute directly to evaluating BRCA1 variants. Furthermore, this strategy will provide valuable insight into diagnoses of other hereditary cancers. However, some points below should be revised before the publication in PLOS Genetics.

General Comments:

1. Distinguishing pathogenic variants from benign variants is important. However, some missense variants cause slight protein conformation change that decreases the HDR activity intermediately and increases the cancer risk moderately. The authors should discuss this in the sections of the Discussion.

2. This study will also contribute to predicting the tumor sensitivity to PARP inhibitors and other DNA-damaging agents. The authors should comment on this in the Introduction and/or Discussion sections.

Specific Comments:

1. In Line 227, a report of the functional assay (Endo et al. Cancer Res Commun. 2021 1(2):90-105) should be cited as a singleton assay.

2. In Line 249, reference number 38 seems to be a mistake.

3. In Figure 6B, indicating the four variants in the graph are better.

Reviewer #2: In this study, the authors have functionally characterized BRCA1 variants of uncertain significance (VUS) using a previously described GFP-based homology-directed DNA repair assay. The authors have focused on residues 2-302 of BRCA1, which includes the RING domain that binds to BARD1. The experimental approach involves changing each codon to all possible other codons using a pool of BRCA1 cDNA with an NNK substitution at each codon. As a result of this substitution, each amino acid is changed to all possible 19 non-WT residues.

The authors have used a new analysis approach to reanalyze previously published results

for BRCA1 residues 2-192. They have modified the method to assess whether the data on the abundance of a variant is sufficient to be included in the analysis without resulting in erroneous classifications of variants. The new approach also differs in how nonfunctional and functional variant are classified. The new approach resulted in an increase in the number of variants that could be classified. In addition, the new approach resulted in increased sensitivity and specificity. In addition, they have generated new variants and determined functional scores for 808 additional variants spanning residues 193-302.

Overall, the study has resulted in functional classification 2172 BRCA1 variants from the N-terminal region of the protein. While the findings are important and relevant, a major concern is the lack of any information on the nucleotide change that resulted in a given amino acid change. This is especially important when comparing their variant classification with ClinVar and SGE data. It is unclear if the authors compared the correct variants (i.e., correct nucleotide change) or compared any variant that resulted in the same amino acid change indicated in ClinVar. It is critical that the authors clarify this. Although in the present study, BRCA1 variants were expressed using a cDNA construct, it is known how silent variants can result in a non-functional protein but affecting splicing. It is equally important that when calculating specificity and sensitivity, variants that have identical nucleotide change should be compared and not identical amino acid change. Authors must include a “nucleotide change” column following HGVS nomenclature system, in Supplementary Tables S1-S5.

Minor comments:

1. Authors have compared their results with SGE data. A number of other CRISPR-based high throughput functional assays have been used to examine the variants in the N-terminal region of BRCA1. Authors should correlate their functional results with these functional assays (e.g., Kweon et al., 2020, Hanna et al, 2021, Sangree et al., 2022)

2. Figure 4, comparing the functional scores of BRCA1 variants in the presence (siControl) and absence (siBRCA1) is an excellent idea. The results are overall very convincing and demonstrate the suitability of the functional assay for classification of BRCA1 VUS. It is unclear why some pathogenic variants behave as benign. Are these synonymous variants that may affect splicing? This effect may be masked because of the use of a cDNA construct.

3. In the text, when mentioning the number of variants that were generated and analyzed in the present study, the authors should mention the total number of variants that were expected to be generated. It is evident from Figure 7 that the number of variants that were generated in the new set was very low. Can the authors comment on why in spite of refining their analysis strategy, the recovery appears to be very low.

4. Figure S3: In 2018, one benign variant was not classified correctly as a benign variant. However, in the present study, all 19 benign variants were correctly classified as a result the specificity went up to 100%. Was this because of the change in the cut-off scores used for considering a variant to be functional or non-functional?

Reviewer #3: The Findlay et al publication of this MAVE was a landmark, presenting evidence of functional impact in 1056 amino acid substitutions in BRCA1. This manuscript presents a new analysis of the MAVE data with a more sensitive methodology, in which the authors can now make function calls for as many as 2172 variants. These new results are largely concordant with the previous ones, and where they differ, the new results seem to be more accurate. This manuscript stands to become a highly impactful publication, with a few modifications.

Major comments:

The comparison to ClinVar does not exclude variants which have been interpreted using functional data. Given the impact of the Findlay et al puiblication, it's highly likely that some of the new ClinVar classifications were made in part with the data from this very assay. At very least, such classifications should be excluded from the comparison. An even better approach would be to exclude all classifications that were made on the basis of functional evidence, since even a different functional assay could introduce circularity.

Minor comments:

Line 110: "removed the variants for which the abundance of variants...". This is awkward wording, using "variants" twice in the same phrase.

Line 148: "all variants" is confusing, as it suggests "each variant". Perhaps it would be better to refer to the overall variant distribution?

Lines 322-326: Awkward wording. This sentence is difficult to parse.

Figure S1: There are three variants circled in red at the right, but the caption does not indicate their significance.

Figure S4: In this image, it's difficult to distinguish between red (LOF: all substitutions) and magenta (LOF: most substitutions)

Finally, while the data is available in GEO, I urge the authors to make these data available also on MaveDB.

**Have all data underlying the figures and results presented in the manuscript been provided?**

Reviewer #1: Yes

Reviewer #2: Yes

Reviewer #3: Yes

PLOS authors have the option to publish the peer review history of their article (what does this mean?). If published, this will include your full peer review and any attached files.

Reviewer #1: No

Reviewer #2: No

Reviewer #3: **Yes: **Melissa S Cline

---

## [Decision Letter · Decision Letter 1]

13 Jul 2023

Dear Dr Parvin,

Thank you very much for submitting your Research Article entitled 'DNA Repair Function Scores for 2172 Variants in the BRCA1 Amino-Terminus' to PLOS Genetics.

The manuscript was fully evaluated at the editorial level and by independent peer reviewers. The reviewers appreciated the attention to an important topic but identified some concerns that we ask you address in a revised manuscript.

We therefore ask you to modify the manuscript according to the review recommendations. Your revisions should address the specific points made by each reviewer.

Yours sincerely,

Anne O'Donnell-Luria, MD, PhD

Academic Editor

PLOS Genetics

David Kwiatkowski

Section Editor

PLOS Genetics

Thank you for your edits. One reviewer has an additional recommendation that I do think is important to add to the manuscript, around the caveat that classifications may have been influenced by the prior functional studies. I appreciate you considering making this additional minor addition to the manuscript.

Reviewer's Responses to Questions

**Comments to the Authors:**

Reviewer #1: The revised manuscript significantly improved, and the authors adequately addressed my previous concerns.

Reviewer #2: The authors have satisfactorily addressed my concerns.

Reviewer #3: This manuscript is a strong, impactful piece of work. The authors have addressed most of the reviewer comments. This reviewer remains concerned that the comparison of the functional assay results to the clinical classifications in ClinVar might be inflated, because some classificaitons might be informed by functional data, and in particular by the results from the Findlay et al publication. But given that many ClinVar classifications aren't explicit about what information infomed the classification, I accept this as a limitation. I suggest that the authors add a brief disclaimer statement saying that it's posslble that some ClinVar classifications could've been informed by the earlier publication on this assay.

**Have all data underlying the figures and results presented in the manuscript been provided?**

Reviewer #1: None

Reviewer #2: Yes

Reviewer #3: Yes

PLOS authors have the option to publish the peer review history of their article (what does this mean?). If published, this will include your full peer review and any attached files.

Reviewer #1: No

Reviewer #2: No

Reviewer #3: **Yes: **Melissa Cline

---

## [Editor Report · Decision Letter 2]

16 Jul 2023

Dear Dr Parvin,

We are pleased to inform you that your manuscript entitled "DNA Repair Function Scores for 2172 Variants in the BRCA1 Amino-Terminus" has been editorially accepted for publication in PLOS Genetics. Congratulations!

Yours sincerely,

Anne O'Donnell-Luria, MD, PhD

Academic Editor

PLOS Genetics

David Kwiatkowski

Section Editor

PLOS Genetics

Comments from the reviewers (if applicable):

**Data Deposition**

http://datadryad.org/submit?journalID=pgenetics&manu=PGENETICS-D-23-00395R2

**Press Queries**

---

## [Editor Report · Acceptance letter]

7 Aug 2023

PGENETICS-D-23-00395R2 

DNA Repair Function Scores for 2172 Variants in the BRCA1 Amino-Terminus 

Dear Dr Parvin, 

We are pleased to inform you that your manuscript entitled "DNA Repair Function Scores for 2172 Variants in the BRCA1 Amino-Terminus" has been formally accepted for publication in PLOS Genetics! Your manuscript is now with our production department and you will be notified of the publication date in due course.

With kind regards,

Zsofi Zombor

PLOS Genetics

On behalf of:
